# Effects of Hydrolysable Tannins as Zinc Oxide Substitutes on Antioxidant Status, Immune Function, Intestinal Morphology, and Digestive Enzyme Activities in Weaned Piglets

**DOI:** 10.3390/ani10050757

**Published:** 2020-04-27

**Authors:** Hansuo Liu, Jiangxu Hu, Shad Mahfuz, Xiangshu Piao

**Affiliations:** State Key laboratory of Animal Nutrition, College of Animal Science and Technology, China Agricultural University, Beijing 100193, China; liuhansuo1115@163.com (H.L.); hujx1007@163.com (J.H.); shadmahfuz@yahoo.com (S.M.)

**Keywords:** antioxidant capacity, immunity, intestinal health, piglets, tannic acid

## Abstract

**Simple Summary:**

Zinc oxide (ZnO) is generally used to control diarrhea and improve gut health in weaned piglets. To protect weaned pigs from intestinal injuries and to decrease environmental zinc load, it is essential to find an alternative to ZnO. In the present study, hydrolysable tannins (HT) showed decreased diarrhea rate and improving gut health via multiple pathways. Herein we demonstrate that HT supplementation may be a potential alternative of ZnO in weaned piglets.

**Abstract:**

Zinc oxide (ZnO) has negative environmental effects and bioavailability in weaned piglets. Thus, finding safe and effective ZnO substitutes to improve intestinal health and to prevent diarrhea of weaned piglets is urgently required. Therefore, this experiment was conducted to evaluate the effects of hydrolysable tannins (HT), ZnO and HT versus ZnO on growth performance, antioxidant status, serum immunity, intestinal morphology, and digestive enzyme activities in weaned pigs. A total of 144 piglets (28 d-old, initial body weight 7.81 ± 0.99 kg) were assigned to 4 treatments with 6 replicates of 6 piglets each. The experiment lasted 28 d (d 1 to 14 as for phase 1 and d 15 to 28 as for phase 2). The dietary treatments include a corn-soybean meal basal diet (CON); ZnO diet (CON + 2000 mg/kg ZnO in phase 1 and 137.5 mg/kg ZnO in phase 2); HT diet (CON + 1000 mg/kg HT in the overall period (d 1 to 28); HT + ZnO diet (CON + 2000 mg/kg ZnO + 1000 mg/kg HT in phase 1, and 137.5 mg/kg ZnO + 1000 mg/kg HT in phase 2). In phase 1, the incidence of diarrhea was lower (*p* < 0.05) in the HT + ZnO group than CON. Serum catalase (CAT) and glutathione peroxidase (GSH-Px) were increased (*p* < 0.01) and malondialdehyde (MDA) was decreased (*p* < 0.01) in the HT + ZnO group than CON. Compared with CON, immunoglobulin M (IgM), immunoglobulin A (IgA) were increased (*p* < 0.05) in the HT + ZnO group. In phase 2, both HT and HT + ZnO had a trend to improve (*p* < 0.10) daily gain. The concentration of total antioxidant capacity (T-AOC) and IgM in serum was higher (*p* < 0.01) in HT compared with CON. Supplementation of HT improved (*p* < 0.01) GSH-Px activities in ileum mucosa than the ZnO group. Compared with CON, trypsin, lipase activities, and villus height of jejunum were improved (*p* < 0.05) in HT and HT + ZnO. The ratio of villus height to crypt depth in the jejunum was improved (*p* < 0.05) in the HT + ZnO group and which also was increased (*p* < 0.05) in ileum in the HT group compared with CON. Propionic acid, butyric acid, and acetic acid concentrations in the colon were increased (*p* < 0.05) in the HT group than CON. Overall, HT + ZnO treatments could be used to replace ZnO for reducing diarrhea and improving antioxidant capacity, immunity, and digestive enzyme activities in weaned piglets.

## 1. Introduction

During the first weeks after weaning, piglets frequently experience poor growth performance and diarrhea due to their immature digestive and immune systems, which led to enormous economic loss in the swine industry [1]. With the phase out of in-feed antibiotics worldwide, zinc oxide (ZnO) is commonly fed at pharmacological doses in weaning piglet diets as a growth promoter and to prevent post-weaning diarrhea [2]. However, on 26 June 2017, the European Commission voted to phase out zinc oxide as a veterinary instrument in the entire union within 5 years, since feeding piglets with high doses of ZnO results in plenty of un-absorbable Zn being released into the environment, increasing the risks of heavy metal contamination [3]. Besides, according to the announcement of the ministry of agriculture of China, the maximum amount of zinc in the compound feed of piglets (≤25 kg) is 110 mg/kg, but the maximum amount of zinc in the feed of piglets is 1600 mg/kg in the first two weeks after weaning. Consequently, the uses of zinc oxide in animal production, lead to a serious and urgent need to find safe and practical alternatives to protect animal health and the environment.

Tannic acid was divided into condensed tannins (CT) and hydrolysable tannins (HT). Condensed tannins have traditionally been considered an anti-nutritional factor as they reduce digestibility and hamper growth performance in monogastric animals [4]. Hydrolysable tannins are heterogeneous groups of natural polyphenolic compounds, with high molecular weight, are water-soluble, and widely found in vegetable feedstuffs and can be extracted from the wood of trees [5]. In monogastric animals, HT have positive effects on the stability of the gastrointestinal microflora and potential antioxidant properties [6]. Previous studies showed that tannins possessed antibacterial, antioxidant, antidiarrheal, and anticancerogenic activities in weaned pigs [7]. Recent studies showed that the application of tannins in diet could improve health status, animal performance, and has a positive role on small intestine morphometric traits in monogastric animals [8,9,10]. Similarly, higher dietary doses of ZnO between 2000 to 4000 mg/kg are generally used to promote growth performance, reduce intestinal permeability, and/ or decrease incidence of diarrhea in weaning piglets [11]. In addition, potential regulatory effects of HT supplementation on intestinal health status in weaned piglets have not been clearly studied before.

Given this background, the purpose of this study hypothesized that HT will support growth and gut health parameters similar to ZnO in a post-weaned piglet diet. Hence, the objective of this experiment was to compare the effects of HT, ZnO, and their combination on growth performance, antioxidant status, immunity, intestinal morphology, and digestive enzyme activities in weaned pigs. 

## 2. Materials and Methods

The experimental protocols used in this experiment were approved by the Institutional Animal Care and Use Committee of China Agricultural University (Beijing, China; No. AW09089103-1). The experiment was carried out at the National Feed Engineering Technology Research Center of the Ministry of Agriculture Feed Industry Center Animal Testing Base (Hebei, China).

### 2.1. Experimental Products

The hydrolysable tannins product was provided by the Gruppo Mauro SilvachimicaSrl (Cuneo, Italy), which was extracted from chestnut wood and contained ≥75% tannin, crude fiber <2.00%, ash <2.50%, and moisture <8.00%.

### 2.2. Experimental Animals and Design

A total of 144 clinically healthy weaned piglets [(Landrace × Yorkshire) × Duroc, 28 d-old, average weight of 7.81 ± 0.99 kg] were randomly allocated to one of four dietary treatments with six replicate pens per treatment (three barrows and three gilts per pen) in a 2 phase feeding program (phase 1: d 1 to 14 and phase 2: d 15 to 28). The dietary treatments were: 1) a corn-soybean meal basal diet (CON); 2) CON + 2000 mg/kg ZnO in phase 1 and 137.5 mg/kg ZnO in phase 2 (ZnO); 3) CON + 1000 mg/kg HT in all phase (HT); 4) CON + 2000 mg/kg ZnO + 1000 mg/kg HT in phase 1, and 137.5 mg/kg ZnO + 1000 mg/kg HT in phase 2 (HT + ZnO All diets were formulated to meet or exceed the nutrient requirements for weaned pigs recommended by National Research Council (NRC, 2012; Table 1) [12] and fed in mash form. ZnO and HT were added in experimental diets and mixed properly via feed millers. The dosage of zinc is added according to the Chinese national standard.

### 2.3. Feeding and Management

All pigs were housed in 2 × 1.2 m pens with plastic slotted floors in an environmentally controlled room. Each pen was equipped with a stainless-steel feeder and a nipple drinker and cleaned daily. Pigs had ad libitum access to feed and water throughout the experiment. Piglets and feed were weighed on d 14 and d 28 to calculate average daily gain (ADG), average daily feed intake (ADFI), and gain to feed ratio (G:F). Piglets were also observed for clinical signs of diarrhea every day.

### 2.4. Detection Index and Measuring Method

All piglets included in the experiment were clinically healthy before the trial began. Fecal scoring was visually assessed every morning and afternoon by observers unaware of the dietary treatments in each pen, and a scoring system was applied to indicate the presence and severity of diarrhea as following: 1 = hard feces; 2 = slightly soft feces; 3 = soft, partially formed feces; 4 = loose, semiliquid feces; and 5 = watery, mucous-like feces [13]. The occurrence of diarrhea was defined as the fecal consistency score for 2 consecutive days at grade 4 or 5, and diarrhea rate was calculated as follows: Diarrhea rate (%) = the total number of diarrheal piglets/(number of piglets × number of days tested) × 100(1)

### 2.5. Sample Collection and Analysis

Samples of all diets were collected at d 1 and d 15. Approximately 200 g of representative fresh feces were collected the last 3 days during both phases. These samples were pooled by pen, dried at 65 °C for 72 h, and ground to pass through a 1 mm sieve (40 mesh) before analysis. Fresh feces were collected on d 14, d 28 from each pen for analysis of volatile fatty acids (VFA).

On d 29, pigs with average body weight (one pig per pen) were humanely killed by exsanguination after electric corona. Segments of the mid-duodenum, mid-jejunum, and mid-ileum were taken and rinsed with cold physiological saline (0.9% saline), then stored in 10% buffered formalin immediately. Tissue samples from the right lobe of the liver were collected, snap-frozen in liquid nitrogen, and stored at −80 °C for further analysis. Mucosal samples from the duodenum, jejunum, and ileum were scraped, rapidly frozen in liquid nitrogen, and stored at −80 °C. The digest from the ileum and cecum were collected and immediately frozen in liquid nitrogen and stored at −80 °C until further assay.

Feed and fecal samples were analyzed for crude protein (CP), dry matter (DM), and ash, according to the methods of AOAC [14]. Gross energy was determined by an automatic isoperibolic oxygen bomb calorimeter (Parr 1281, Automatic Energy Analyzer; Moline, IL, USA). The chromium content in the diets and feces was measured using an atomic absorption spectrophotometer (Z-5000; Hitachi, Tokyo, Japan) according to the procedure of Williams et al. [15]. Organic matter (OM) was calculated as 1 − Ash content (DM-basis). Nutrient digestibility was determined by the equation as follows:Apparent total tract digestibility_nutrient_ (ATTD) = 1 − (Cr_diet_ × Nutrient_feces_)/(Cr_feces_ × Nutrient_diet_)(2)
where Cr_diet_ represents chromium concentration in the in the diet [g/kg of DM], Nutrient_feces_ is the concentration of nutrients or other component in the ileal digesta or feces [g/kg of DM], Cr_feces_ represents chromium concentration ileal digesta or feces [g/kg of DM], and Nutrient_diet_ is the concentration of nutrients or other component in the diets [g/kg of DM].

On the morning of d 14 and d 28, blood samples (8 mL) were collected from 1 pig [(median body weight (BW)] in each pen via jugular vein puncture into additive-free vacutainer tubes (Becton Dickinson Vacutainer Systems, Franklin Lakes, NJ, USA). After 3 h, blood samples were centrifuged at 2000 r/min for 10 min at 4 °C to recover serum, which was stored at −20 °C until analysis. Immunoglobulins (IgA, IgG, and IgM) were measured by an ELISA kit (IgA, IgG, and IgM quantitation kit; Bethyl Laboratories, Inc., Montgomery, TX, USA). Determination of serum total antioxidant capacity (T-AOC), total superoxide dismutase (T-SOD), glutathione peroxidase (GSH-PX), and malondialdehyde (MAD) levels were conducted by spectrophotometric methods using a spectrophotometer (LengGuang SFZ1606017568, Shanghai, China) following the instructions of the kit’s manufacturer (Nanjing Jiancheng Bioengineering Institute, Nanjing, China).

At the end of the experiment, one barrow closest to the average pen body weight was selected from each pen to collect liver samples for antioxidant and immunity indices, aseptic ileum samples for morphology test, chyme in the cecum, and the colon for volatile fatty acids (VFA) content test. The histological samples were rapidly fixed in neutral buffered formalin. The sections of small intestine were excised, dehydrated, and embedded in paraffin wax before 4 transverse sections (3 to 5 μm) were cut, installed on glass slides, and stained with hematoxylin and eosin. Five non-successive sections from each tissue sample were selected and measured. The height of at least 10 randomly orientated villi and their adjoining crypts were determined with a light microscope (CK-40, Olympus, Tokyo, Japan) at 40× magnification and analyzed with an image analyzer (Lucia Software. Lucia, ZaDrahou, Czechoslovakia).

For the analysis of digestive enzyme activities, the digesta of the gastrointestinal tract was thawed and homogenized in 4 volumes of ice-cold 0.9% sodium chloride solution. The homogenate was centrifuged at 13,800× *g* for 20 min at 4 °C and the supernatant was analyzed for amylase, trypsin, and lipase activities. All enzyme activities were measured spectrophotometrically (PU 8720 UV/VIS Scanning Spectrophotometer, Pye Unicam, Cambridge, UK) described by Lee et al. [16].

As for VFA content test, frozen samples were thawed at room temperature and approximately 1.5 g of sample was placed into a centrifuge tube, and mixed with 1.5 mL of sterile water. The content was thoroughly mixed and centrifuged at 15,000 r/min for 10 min at 4 °C. The supernatant (1 mL) was transferred into a gas chromatograph sample bottle, and 200 µL meta-phosphoric acid was added. The bottle was immersed in ice for 30 min and then centrifuged at 15,000× *g* for 10 min at 4 °C. A Hewlett Packard 5890 gas chromatograph (HP, Avondale, PA, USA) was used to determine VFA concentrations.

### 2.6. Statistical Analysis

All data were analyzed using the mixed procedure of SAS (version 9.2, SAS Inst. Inc., Cary, NC). Dietary treatment was fixed effect, sex and body weight were random effects. For the performance data and fecal score, pen was considered as the experimental unit. For the other data, individual pig was treated as the experimental unit. Statistical differences among treatments were separated by Student–Neuman–Keul’s multiple range tests. Significance was designated at *p* ≤ 0.05, while a tendency for significance was designated at 0.05 < *p* ≤ 0.10.

## 3. Results

### 3.1. Growth Performance and Diarrhea Rate of Piglets

The results of the growth performance and the diarrhea rate of weaned pigs were presented in Table 2. Supplementation of ZnO and HT reduced (*p* < 0.01) the diarrhea rate in phase 1. Among the groups, no differences were observed on ADFI and G:F in the current study. Compared with CON, HT + ZnO tended to increase (*p* < 0.10) ADG during phase 2 and the overall period.

### 3.2. The ATTD of Nutrients

The results observed regarding nutrient digestibility for different diets did not show differences (*p* > 0.01) during phase 1. During phase 2, piglets in the HT + ZnO group expressed greater (*p* < 0.01) ATTD of DM, OM, CP, and GE compared with CON (Table 3).

### 3.3. Antioxidant and Immunity Parameters in Serum, Intestinal Mucosal, and Liver

Antioxidant and immune indexes in the serum of piglets were showed in Table 4. On d 14, serum CAT and GSH-Px concentrations were increased (*p* < 0.01) in HT and HT + ZnO fed groups compared with CON. Moreover, T-AOC was increased (*p* < 0.01) in the HT + ZnO group than the other groups. In addition, MDA was decreased (*p* < 0.01) in the HT + ZnO fed group than the other groups. On d 28, CAT and T-AOC was increased (*p* < 0.01) in the HT + ZnO fed group than CON. Serum GSH-Px was increased (*p* < 0.01) and MDA was decreased (*p* < 0.01) in the test groups than CON. Besides, IgM was increased (*p* < 0.01) in HT and HT + ZnO fed groups in all phases than CON. In addition, compared with CON, HT and HT + ZnO fed groups tended to increase (*p* < 0.10) IgG and IgA was increased (*p* < 0.05) in HT + ZnO fed groups on d 14. Compared to CON, serum IgG was increased (*p* < 0.05) in the HT + ZnO group on d 28. During all phases, no differences in antioxidant and immune parameters in serum between HT and ZnO treatments were observed.

The effect of HT and ZnO supplementation on antioxidant enzymes in intestinal mucosa was presented in Table 5. Compared with CON, T-AOC, GSH-Px were increased (*p* < 0.01) in the HT + ZnO fed group. Additionally, CAT in duodenum mucosa tended to increase (*p* < 0.1) in the HT + ZnO fed group than CON. Serum T-SOD was found higher (*p* < 0.01) in HT and HT + ZnO fed groups than CON. GSH-Px was higher (*p* < 0.05) and MDA was decreased (*p* < 0.05) of ileum mucosa in the HT fed group than ZnO. Among the treatments, no significance difference (*p* > 0.1) was observed on antioxidant indices of liver in the current study.

### 3.4. Intestinal Morphology and Digestive Enzyme Activities

The α-AMY activities of the duodenum were increased (*p* < 0.01) in the HT + ZnO fed group than CON and ZnO (Table 6). Trypsin and lipase activities of the jejunum were improved (*p*< 0.01) in HT and HT + ZnO fed groups than in CON. In addition, α-AMY activities of the jejunum were improved (*p* < 0.05) in HT and HT + ZnO fed groups than ZnO. Lipase of ileum was improved (*p* < 0.01) in HT and HT + ZnO fed groups than CON. 

Villus height of jejunum was increased (*p* < 0.01) in HT and HT + ZnO fed groups than CON (Table 7). Villus height of ileum tended to improve (*p* < 0.10) in HT and HT + ZnO fed groups than CON. Among the treatments, no differences (*p* > 0.10) were observed on crypt depth in the current study. The ratio of villus height to crypt depth of ileum was increased (*p* < 0.05) in the HT fed group than CON. Furthermore, the ratio of villus height to crypt depth in the jejunum was increased (*p* < 0.05) in the HT + ZnO fed group than CON.

### 3.5. Volatile Fatty Acids

The concentration of VFA in the cecum did not differ among treatments (Table 8). However, propionic acid and butyric acid concentrations in the colon were increased (*p* < 0.05) in the HT fed group than CON. Acetic acid in the colon was improved (*p* < 0.05) in HT and HT + ZnO fed groups than CON and ZnO. Total VFA in the colon was improved (*p* < 0.01) in HT and HT + ZnO fed groups than CON.

## 4. Discussion

In the present research, results showed that hydrolysable tannins (HT) supplementation had no negative influence on feed efficiency. Čandek-Potokar et al. [17] reported that there were no significant effects on average daily gain (ADG) and gain to feed ratio (G:F) in pigs fed with tannins supplemented diets. Myrie et al. [18] reported that dietary supplementation of tannins at 15 g/kg showed no effect on growth performance in weaned pigs. Similar findings are also shown by Zhao et al. and García et al., who reported the dietary supplementation of tannins have no effect on growth performance in sheep [19,20]. However, some results revealed that several tannin sources could improve performance in pigs and chickens [8,9]. In the previous studies, the application of lower doses of tannin (1130, 2250, or 4500 mg/kg) had positive effect on ADG, ADFI, and G:F of weaned pigs in comparison to the control diet [7]. In our study, we found pigs fed with HT had a similar effect on ADG compared to the ZnO treatment, although there was insignificant difference with CON. The possible reason was that the positive effects of tannins improve the intestinal health status through their antioxidant, anti-inflammatory, and anti-microbial activities [21].

Diarrhea is a severe problem which is related to a higher mortality rate in early weaned pigs. Early weaned pigs are susceptible to a high incidence of gastrointestinal disturbances with diarrhea, due to multi-factorial stressors which include changes in nutrition, surrounding environment, microbial challenges, and social relations [22]. Previous studies reported that inclusion of pharmacological doses of ZnO in weaned pig diets not only improved the growth performance but also decreased diarrhea incidence [2]. In this research, dietary supplementation with HT showed a reduced diarrhea rate as effective as ZnO treatment. The reason was due to tannins being able to prevent some digestive disorders and leakage of liquids from body tissues into intestine lumen and prevent diarrhea [23]. Another possible reason was that tannins in the digestive tract could form a thin film in the intestine walls, therefore reducing the intestinal concentrations of the stimulation and exerting antidiarrheal effect [24]. Furthermore, in all stages, IgM was increased in pigs fed with the HT diet, which could be necessary and beneficial to prevent pigs diarrhea. Our data demonstrated that HT has a strong anti-diarrhea effect even at the dose of 1000 mg/kg in diets, indicating that HT has the potential to replace ZnO as an anti-diarrhea agent. 

A great deal of research on pig nutrition has focused on the utilization of different feed grains rich in tannins, but little work has been carried out using purified tannin as feed additives till now. Hydrolysable tannins are known to be degraded and/or absorbed in the gastrointestinal tract. In this study, a mixture of HT and ZnO improved apparent nutrient digestibility of DM, OM, CP, and GE than other groups during the last period. Data are scarce regarding the effects of tannins on nutrient digestibility in pigs. Our results showed that the apparent nutrient digestibility has no effect on pigs only fed with the HT single. However, Schiavone et al. [25] suggested that HT improved the utilization efficiency of DM in broilers. On the one hand, the improvement in nutrient absorption may be partly explained by stimulating secretions of saliva, bile, and enhanced enzyme activity induced by tannins [26]. Moreover, the inhibitory and protective function of hydrolysable tannins has been proven to protect the intestinal mucosa from pathogens and harmful bacteria [27]. Whereas, pigs fed with HT having no effect on the nutrition utilization efficiency in our experiment may be associated with the animal species, and the activity and sources of tannins.

To further figure out the underlying mechanism of nutrient digestibility with ZnO, HT, and HT + ZnO diets compared with CON, index reflecting intestinal health and digestive enzymes activities of weaned pigs were in this study. Vallee and Falchuk [28] reported that zinc played an important role in the synthesis of enzymes, which is a cofactor of more than 300 metallo enzymes and essential for nutrient. In the present study, results showed that trypsin, lipase, and α-AMY activities in different intestinal segments were improved in HT and HT + ZnO compared with other groups. We hypothesized that there may be a synergistic effect between HT and ZnO, which leads to enhance beneficial functions. Moreover, tannins could positively affect digestive enzymes activity, microbial fermentation in the gut, and decrease concentrations of stress-related enzymes in plasma [26].

Intestinal villi size is important for nutritional digestion and absorption as longer villi possess a larger surface area for absorption [29]. Bilić-Šobot et al. [10] showed that supplementation of HT increased villus height and villus perimeter of the duodenum. However, we found dietary supplementation HT increased villus height of the jejunum and tended to improve the villus height of the ileum, suggesting that tannins may be absorbed in a specific intestine segment. At weaning, multi-factorial stressors can trigger crypt hyperplasia, leading to reduced surface area for nutrient absorption, and ultimately poor performance and diarrhea [30]. Biagi et al. [7] showed that the depth of ileum crypts tended to decrease in pigs fed tannins at 2.25 and 4.5 g/kg. In contrast, the present study showed that the crypt depth of the small intestine was not affected by tannin supplementation. Whereas, the villus height to crypt depth ratio in the jejunum and ileum was increased by the dosage of HT. Similarly, Fiesel et al. [31] stated that pigs fed with grape seed and grape marc meal extract as a supplement showed a higher villus height to crypt depth ratio in the duodenum than control diets, which may be due to tannins possessing selective antimicrobial properties, inhibiting extracellular microbial enzymes, and modulating metabolic oxidative phosphorylation. In addition, tannins modifying cell wall morphology and increasing membrane permeability was due to the bind to microbial cell membranes [21]. A previous study showed that plant polyphenols have shown to be effective in inhibiting the reproduction of harmful microorganisms, regulating the intestinal microflora, and thereby reducing the toxins produced by the proliferation of harmful microorganisms, thus alleviating intestinal morphological damage [32]. After weaning, the morphological damage of intestinal villi in piglets may be related to oxidative stress [33]. Free radicals produced by oxidative stress can induce damage in intestinal epithelial cells. Studies have shown that intestinal epithelial cells move from the base of the crypt to the end of the villi and form new villi cells with absorption capacity to supplement normal villi cells apoptosis [34]. Therefore, HT can reduce the oxidative stress of the intestine during weaning, decrease the damage of free radicals to epithelial cells, and thereby protect the intestinal morphology. Accordingly, one possible explanation for the improved digestibility enzyme activities may be attributed to tannins protecting intestinal morphology health and reducing the incidence of intestinal disease. In addition, because tannins are able to form complexes with proteins, it is not surprising that they also bind to enzymes; this has implications, for tannins may improve enzymes activity [35]. 

The complex system of the antioxidant enzymes system was the first defense to protect the organism against harmful peroxidation [36]. Hagerman et al. [37] reported that CT and HT of relatively high molecular weight exhibited greater antioxidant activities than simple phenolics. The present study showed that a diet including HT may alleviate the oxidative stress by enhancing the antioxidant enzyme activities in serum and small intestine, which was reflected by improved GSH-Px activities in the ileum and duodenum. Similar reports were also observed in broilers by Liu et al. [38]. The possible reason for these findings is that HT can selectively induce antioxidant enzyme gene expression via modulating redox-sensitive signaling pathways by inhibiting lipid peroxidation and quenching the oxygen free radicals in the gut [39]. Higher serum antioxidant enzymes concentrations were associated with the improvement of intestinal morphology in the HT and HT + ZnO diet.

The increased concentrations of immunoglobulin in pigs could improve the immune function and help pigs develop their own immune system, thus alleviating the weaning stress [40]. Immunoglobulin A is the major antibody involved in the mucosal immunity, which neutralizes intracellular pathogens by intimate cooperation with an innate nonspecific defense system. In the present research, serum IgA was increased in pigs fed with HT as well as ZnO, which can prevent pathogens from damaging the gut wall, and thereby maintain structural and functional integrity of the small intestine [41]. Furthermore, IgM is another primary antibody produced at the initial stage of antibody response. A serum IgM increase in the HT diet could be necessary and beneficial for pigs since they are very susceptible to stressors including diseases and digestive disorders during the first weeks of post-weaning [42]. Kong et al. [43] reported that propolis flavone could promote lymphocyte proliferation secrete IL-2 and IL-4, increase the contents of IgG, IgA, and IgM in the blood, and improve the immune function of piglets. We speculate that the improvement of immune function in pigs was related to performance and intestinal health, thereby modifying cell wall morphology and eventually improving immune function in piglets. A previous study also showed that immunity was influenced by improving the antioxidant status of the animal [44]. Low feed intake and starvation of piglets during the first few days after weaning represent a major challenge for producers. Therefore, 1000 mg/kg HT supplementation during this period probably will prevent oxidative stress and improve immune status by increasing serum IgA and IgM content.

Volatile fatty acids (VFA), mainly including acetate, propionate, and butyrate are the main end products of fermentation of indigestible carbohydrates by microbiota in the large intestine. Biagi et al. [7] reported that increased concentrations of tannins linearly reduced total gas production and concentrations of acetic, propionic, and n-butyric acid. Nevertheless, in the present study, butyric acid and propionate were increased in the colon with the HT diet, indicating that the availability of indigestible carbohydrates was increased in the colon of pigs. Molino et al. [45] observed that a high prebiotic activity may be beneficial of nutrient digestibility because chestnut hydrolysable tannins are easily available for microbial fermentation. Dietary HT supplementation has different results that may be due to an indigestible nutritional ingredient might have been fermented in the large intestine by HT supplementation instead of utilized in the small intestine. 

## 5. Conclusions

The addition of HT improved intestinal morphology, and digestive enzyme activities in weaned piglets. The results further showed that dietary HT supplementation improved the serum and intestinal mucosa antioxidant capacity, immune function, and VFA in weaned pigs. Further study is needed to study the effects of tannin on intestinal microorganisms and the optimal replacement dose as a substitute for zinc oxide in weaned piglets.

## Figures and Tables

**Table 1 animals-10-00757-t001:** Composition and nutrient levels of basal diets (%, as-fed basis).

Ingredients	D 1 to 14	D 15 to 28
Corn	60.02	63.00
Soybean meal, 43%	10.00	15.50
Extruded soybean	12.00	8.00
Spray dried plasma protein	3.00	0.00
Fish meal, 64.6%	5.00	4.00
Whey powder, 3.8%	4.00	4.00
Soybean oil	2.76	2.27
Dicalcium phosphate	0.67	0.66
Limestone	0.96	0.80
Salt	0.30	0.30
L-lysine HCl, 78%	0.36	0.45
DL-Methionine, 98%	0.08	0.08
L-Threonine, 98%	0.08	0.15
L-Tryptophan, 98%	0.02	0.04
Chromic oxide	0.25	0.25
Vitamin-mineral premix ^1^	0.50	0.50
Calculated nutrient levels ^2^		
Digestible energy, MJ/kg	14.41	14.07
Crude protein	19.28	17.21
Calcium	0.80	0.70
Digestible phosphorus	0.40	0.33
Standardized ileal digestible lysine	1.44	1.47
Standardized ileal digestible methionine	0.45	0.43
Standardized ileal digestible threonine	0.93	0.86
Standardized ileal digestible tryptophan	0.22	0.20

Note: ^1^ Vitamin and mineral premix provided the following per kilogram of diet: 12,000 IU vitamin A as vitamin A acetate, 2500 IU vitamin D as vitamin D_3_, 30 IU vitamin E as DL-α-tocopheryl acetate, 12 μg of vitamin B_12_, 3 mg vitamin K as menadione sodium bisulfate, 15 mg D-pantothenic acid as calcium pantothenate, 40 mg of nicotinic acid, 400 mg choline as choline chloride, 30 mg Mn as manganese oxide, 80 mg Zn as zinc oxide, 90 mg Fe as iron sulfate, 10 mg Cu as copper sulfate, 0.35 mg I as ethylenediamine dihydroiodide, and 0.3 mg Se as sodium selenite. ^2^ Digestible energy, crude protein, digestible lysine, digestible methionine, digestible threonine were analyzed values. Other values were calculated.

**Table 2 animals-10-00757-t002:** Effects of hydrolyzed tannic acid (HT) and zinc oxide (ZnO) individual supplementation or combination on growth performance and diarrhea rate of weaned pigs.

Item	CON ^1^	ZnO ^1^	HT ^1^	HT + ZnO ^1^	SEM ^2^	*p*-Value
d 1 BW, kg	7.81	7.80	7.81	7.81	0.01	0.91
d 14 BW, kg	12.84	12.67	12.97	13.37	0.18	0.09
d 28 BW, kg	17.70	17.75	17.95	18.56	0.33	0.27
d 1 to14						
Average daily gain, g	333	335	354	359	10.75	0.26
Average daily feed intake, g	469	470	491	514	16.41	0.21
Gain: Feed	0.71	0.71	0.72	0.70	0.02	0.86
Diarrhea rate	9.33 ^a^	3.00 ^b^	3.50 ^b^	0.83 ^b^	0.01	<0.01
d 15 to 28						
Average daily gain, g	361	378	394	406	11.86	0.08
Average daily feed intake, g	700	695	746	751	30.42	0.44
Gain: Feed	0.54	0.54	0.53	0.55	0.03	0.98
d 1 to 28						
Average daily gain, g	347	356	374	382	9.18	0.06
Average daily feed intake, g	585	583	619	633	20.40	0.26
Gain: Feed	0.64	0.61	0.62	0.61	0.03	0.89

^1^ CON, the control; ZnO, zinc oxide; HT, hydrolyzed tannic acid; HT + ZnO, the mixture of zinc oxide. ^2^ SEM means standard error of the mean. ^a, b^ different superscripts within a row indicate a significant difference (*p* < 0.05).

**Table 3 animals-10-00757-t003:** Effect of HT and ZnO individual supplementation or combination on apparent total tract digestibility of weaned pigs (%).

Item	CON ^1^	ZnO ^1^	HT ^1^	HT + ZnO ^1^	SEM ^2^	*p*-Value
d 14						
Dry matter	83.96	84.75	84.37	82.89	0.67	0.27
Organic matter	86.88	87.16	87.12	85.85	0.58	0.37
Crude protein	77.40	77.96	78.93	76.28	1.15	0.45
Gross energy	83.30	84.51	84.29	82.28	0.69	0.13
d 28						
Dry matter	84.49 ^b^	84.12 ^b^	84.66 ^b^	86.63 ^a^	0.39	<0.01
Organic matter	87.24 ^b^	87.00 ^b^	87.59 ^b^	89.16 ^a^	0.35	<0.01
Crude protein	77.05 ^b^	78.46 ^b^	79.82 ^ab^	82.10 ^a^	0.83	<0.01
Gross energy	84.30 ^b^	83.96 ^b^	84.51 ^b^	86.29 ^a^	0.40	<0.01

^1^ CON, the control; ZnO, zinc oxide; HT, hydrolyzed tannic acid; HT + ZnO, the mixture of zinc oxide and hydrolyzed tannic acid. ^2^ SEM means standard error of the mean. ^a, b^ Different superscripts within a row indicate a significant difference (*p* < 0.05).

**Table 4 animals-10-00757-t004:** Effect of HT and ZnO individual supplementation or combination on serum antioxidant and immune indexes of weaned pigs.

Item ^1^	CON ^2^	ZnO ^2^	HT ^2^	HT + ZnO ^2^	SEM ^3^	*p*-Value
d 14						
CAT(U/mL)	24.36 ^c^	29.18 ^bc^	33.19 ^b^	47.96 ^a^	1.17	<0.01
GSH-Px (U/mL)	684 ^c^	855 ^b^	919 ^b^	1049 ^a^	24.80	<0.01
SOD (U/mL)	71.72 ^ab^	66.14 ^ab^	64.16 ^b^	73.95 ^a^	2.21	0.03
T-AOC (U/mL)	13.31 ^ab^	12.80 ^ab^	12.78 ^b^	15.90 ^a^	0.97	0.02
MDA (nmol/mL)	5.48 ^a^	4.84 ^ab^	4.63 ^b^	2.71 ^c^	0.20	<0.01
IgA (g/L)	1.16 ^b^	1.38 ^ab^	1.42 ^ab^	1.60 ^a^	0.09	0.03
IgM (g/L)	3.13 ^c^	3.38 ^bc^	3.48 ^b^	3.96 ^a^	0.06	<0.01
IgG (g/L)	20.67	21.25	21.56	23.11	0.62	0.08
d 28						
CAT (U/mL)	22.97 ^b^	27.72 ^b^	31.15 ^ab^	39.47 ^a^	2.73	0.01
GSH-Px (U/mL)	666 ^b^	806 ^a^	850 ^a^	914 ^a^	28.48	<0.01
SOD (U/mL)	49.05 ^b^	68.24 ^a^	65.06 ^ab^	49.08 ^b^	4.38	0.02
T-AOC (U/mL)	8.17 ^c^	9.64 ^bc^	11.89 ^ab^	14.19 ^a^	0.55	<0.01
MDA (nmol/mL)	5.69 ^a^	5.21 ^a^	4.94 ^a^	3.82 ^b^	0.24	<0.01
IgA (g/L)	1.18 ^b^	1.31 ^ab^	1.36 ^ab^	1.47 ^a^	0.07	0.07
IgM (g/L)	3.02 ^c^	3.20 ^bc^	3.28 ^b^	3.87 ^a^	0.05	<0.01
IgG (g/L)	20.01 ^b^	20.21 ^b^	20.64 ^ab^	22.56 ^a^	0.52	0.02

^1^ T-AOC = total antioxidant capacity; MDA = malondialdehyde; CAT = catalase; SOD = superoxide dismutase; GSH-Px = glutathione peroxidase; Ig = immunoglobulins. ^2^ CON, the control; ZnO, zinc oxide; HT, hydrolyzed tannic acid; HT + ZnO, the mixture of zinc oxide and hydrolyzed tannic acid. ^3^ SEM means standard error of the mean. ^a–c^ Different superscripts within a row indicate a significant difference (*p* < 0.05).

**Table 5 animals-10-00757-t005:** Effects of HT and ZnO individual supplementation or combination on serum antioxidant capacity of weaned piglets.

Item ^1^	CON ^2^	ZnO ^2^	HT ^2^	HT + ZnO ^2^	SEM ^3^	*p*-Value
Duodenum						
T-AOC (U/mg)	6.90 ^b^	8.72 ^ab^	8.90 ^ab^	10.70 ^a^	0.55	<0.01
CAT (U/mg)	3.67	4.17	5.02	5.51	0.42	0.05
GSH-Px (U/mg)	78.98 ^b^	83.04 ^b^	89.06 ^ab^	116.72 ^a^	6.57	0.01
SOD (U/mg)	6.41 ^b^	6.39 ^b^	7.80 ^a^	8.48 ^a^	0.29	<0.01
MDA (nmol/mg)	0.61 ^a^	0.57 ^ab^	0.55 ^ab^	0.46 ^b^	0.03	0.05
Jejunum						
T-AOC (U/mg)	8.62	8.12	7.11	7.20	0.23	0.22
CAT (U/mg)	2.66	2.35	2.87	2.90	3.53	0.37
GSH-Px (U/mg)	61.56	47.58	51.38	63.64	0.21	0.07
SOD (U/mg)	6.12	5.70	6.16	6.42	0.55	0.17
MDA (nmol/mg)	0.66	0.79	0.74	0.70	0.03	0.07
Ileum						
T-AOC (U/mg)	8.18	8.54	8.51	6.54	0.81	0.29
CAT (U/mg)	4.16	4.49	4.49	4.28	0.42	0.82
GSH-Px (U/mg)	77.71 ^ab^	69.71 ^b^	94.13 ^a^	79.30 ^ab^	4.66	0.03
SOD (U/mg)	7.37	6.80	7.57	6.87	0.38	0.43
MDA (nmol/mg)	0.58 ^ab^	0.63 ^a^	0.53 ^b^	0.59 ^ab^	0.02	0.03
Liver						
CAT (U/mg)	5.21 ^b^	7.17 ^a^	6.04 ^ab^	5.49 ^b^	0.27	<0.01
GSH-Px (U/mg)	71.56	87.26	86.16	77.79	6.19	0.32
SOD (U/mg)	7.06 ^b^	9.09 ^a^	8.60 ^ab^	8.01 ^ab^	0.36	0.03
MDA (nmol/mg)	0.62	0.56	0.57	0.59	0.02	0.36

^1^ T-AOC = total antioxidant capacity; MDA = malondialdehyde; CAT = catalase; SOD = superoxide dismutase; GSH-Px = glutathione peroxidase; Ig = immunoglobulins. ^2^ CON, the control; ZnO, zinc oxide; HT, hydrolyzed tannic acid; HT + ZnO, the mixture of zinc oxide and hydrolyzed tannic acid. ^3^ SEM means standard error of the mean. ^a–c^ Different superscripts within a row indicate a significant difference (*p* < 0.05).

**Table 6 animals-10-00757-t006:** Effects of HT and ZnO individual supplementation or combination on intestinal enzymes activities of weaned pigs.

Item	CON ^1^	ZnO ^1^	HT ^1^	HT + ZnO ^1^	SEM ^2^	*p*-Value
Duodenum						
Trypsin (U/mg)	16.40	15.45	14.71	14.21	0.98	0.46
Lipase (U/mg)	16.29	15.73	14.99	14.63	0.49	0.14
α-AMY (U/g)	291 ^b^	348 ^b^	397 ^ab^	483 ^a^	25.53	<0.01
Jejunum						
Trypsin (U/mg)	12.67 ^b^	15.90 ^ab^	16.67 ^a^	17.23 ^a^	0.74	<0.01
Lipase (U/mg)	11.96 ^b^	15.67 ^a^	16.61 ^a^	17.51 ^a^	0.80	<0.01
α-AMY (U/g)	151 ^b^	174 ^b^	223 ^a^	264 ^a^	10.54	0.01
Ileum						
Trypsin (U/mg)	15.90 ^b^	19.61 ^a^	15.33 ^b^	16.57 ^ab^	0.73	0.01
Lipase (U/mg)	15.80 ^c^	20.61 ^b^	22.93 ^ab^	24.31 ^a^	0.63	<0.01
α-AMY (U/g)	79.87	141.58	124.73	24.34	38.91	0.21

^1^ CON, the control; ZnO, zinc oxide; HT, hydrolyzed tannic acid; HT + ZnO, the mixture of zinc oxide and hydrolyzed tannic acid. ^2^ SEM means standard error of the mean. ^a, b^ Different superscripts within a row indicate a significant difference (*p* < 0.05).

**Table 7 animals-10-00757-t007:** Effects of HT and ZnO individual supplementation or combination on intestinal morphology of weaned pigs.

Item	CON ^1^	ZnO ^1^	HT ^1^	HT + ZnO ^1^	SEM ^2^	*p*-Value
Duodenum						
Villus height (μm)	462	570	538	557	37.21	0.24
Crypt depth (μm)	285	349	294	306	17.23	0.11
Villus height/Crypt depth	1.64	1.63	1.85	1.82	0.12	0.45
Jejunum						
Villus height (μm)	424 ^b^	517 ^ab^	544 ^a^	545 ^a^	24.67	0.02
Crypt depth (μm)	265	294	279	272	12.92	0.47
Villus height/Crypt depth	1.60 ^b^	1.77 ^ab^	1.97 ^ab^	2.01 ^a^	0.09	0.02
Ileum						
Villus height (μm)	365	406	440	403	17.94	0.09
Crypt depth (μm)	241	238	232	227	9.28	0.71
Villus height/Crypt depth	1.51 ^b^	1.73 ^ab^	1.93 ^a^	1.78 ^ab^	0.09	0.04

^1^ CON, the control; ZnO, zinc oxide; HT, hydrolyzed tannic acid; HT + ZnO, the mixture of zinc oxide and hydrolyzed tannic acid. ^2^ SEM means standard error of the mean. ^a, b^ Different superscripts within a row indicate a significant difference (*p* < 0.05).

**Table 8 animals-10-00757-t008:** Effects of HT and ZnO individual supplementation or combination on volatile fatty acids (VFA) contents in the colon and cecum of weaned pigs.

Item	CON ^1^	ZnO ^1^	HT ^1^	HT + ZnO ^1^	SEM ^2^	*p*-Value
Cecum						
Acetic acid	3.33	3.89	3.89	4.03	0.19	0.10
Propionic acid	2.42	2.59	2.49	2.63	0.22	0.90
Isobutyric acid	0.04	0.10	0.13	0.10	0.03	0.20
Butyric acid	1.03	1.17	1.21	1.15	0.13	0.78
Isovaleric acid	0.18	0.13	0.11	0.11	0.03	0.37
Valeric acid	0.20	0.31	0.18	0.27	0.06	0.46
Total volatile fatty acid	7.20	8.19	8.01	8.29	0.46	0.37
Colon						
Acetic acid	3.41 ^b^	3.40 ^b^	4.26 ^a^	4.27 ^a^	0.23	0.03
Propionic acid	2.19 ^b^	2.27 ^ab^	3.08 ^a^	2.91 ^ab^	0.19	0.02
Isobutyric acid	0.11	0.10	0.09	0.17	0.03	0.19
Butyric acid	0.98 ^b^	1.15 ^b^	1.77 ^a^	1.28 ^ab^	0.13	0.01
Isovaleric acid	0.11	0.11	0.13	0.13	0.01	0.24
Valeric acid	0.21	0.36	0.39	0.37	0.08	0.43
Total volatile fatty acid	7.00 ^c^	7.40 ^bc^	9.72 ^a^	9.13 ^ab^	0.44	<0.01

^1^ CON, the control; ZnO, zinc oxide; HT, hydrolyzed tannic acid; HT + ZnO, the mixture of zinc oxide and hydrolyzed tannic acid. ^2^ SEM means standard error of the mean. ^a, b^ Different superscripts within a row indicate a significant difference (*p* < 0.05).

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
