# Peer review of "Effects of Hydrolysable Tannins as Zinc Oxide Substitutes on Antioxidant Status, Immune Function, Intestinal Morphology, and Digestive Enzyme Activities in Weaned Piglets"

_animals, 2020, doi:10.3390/ani10050757_

Round 1
Reviewer 1 Report
The authors have investigated the potential of HT as alternative to the use of ZnO. The results illustrate that it is worthwhile investigation this further. It is advisable to restrict the conclusions to the real statistically significant differences they observed (and not mention the tendencies) since sample sizes and the experimental set-up warrant some prudency in this regard.
Some detailed comments:
Simple summary
Line 14 …. although inorganic zinc could not be efficiently absorbed and retained in piglets … I do not see the relevance of mentioning inorganic zinc in the simple summary
Line: 17 …‘positive effects on decreasing diarrhea rate’ … replace by … ‘decreased diarrhea rate’…
Line 17 … ‘in terms of’ … replace by … ‘via‘…
Abstract
Line 20 … low bioavailable … replace by bioavailability
Line 23 evaluate the effects of hydrolysable tannins (HT) and ZnO on … suggest that you give the piglets a combination of HT and ZnO … would the use of HT versus ZnO not be more appropriate
Line 25: given that weaning is carried out at different ages and that age can have an effect on gut health parameters can you also indicate the approximate ages of the 144 piglets used?
Line 32: never start a sentence with ‘And’
Introduction:
Information on the guidelines on the use of ZnO and the ban in the EU could be mentioned as additional reasons to investigate alternatives.
The authors provide details and references on the effect of HT’s which seemingly are similar to those of ZnO. It would perhaps be useful to review in short, the modes of action of ZnO as well.
In the experimental set-up, two phases are included as is a combination between HT’s and ZnO .. can the authors provide reasons for this setup? Why two phases with different concentrations, why investigate the combination between HT+ZnO?
Ideally the introduction is concluded by detailed hypotheses. The authors do not compare HT with ZnO … since one of the treatments includes a combination of HT and ZnO.
Material and methods
In view of easy comparison could levels of zinc oxide also be given as ppm ?
The experimental design included barrows and gilts. Sex was included in the statistical analysis, however some samples were only taken from barrows? Thus for which parameters sex was included, and for which parameters was it not considered?
Was fecal scoring carried out at pen level?
Blood samples and tissue samples were taken from 1 pig per pen/per treatment/per replicate. How was sex taken into account (given that for eg. volatile fatty acids it is explicitally mentioned samples were taken from a barrow)?
Line 119 and line 137 mention taking blood samples. I would only mention it in line 137 since it is followed by details on how the sample was treated after taking.
Line 152 what do the authors mention with ‘small caps’?
In how many sections were villi and crypts measured? What magnification was used to estimated the heights and depths?
Results:
Pay attention to having superscripts in Table 3 (some a’s and b’s are not in superscript).
Why was the ATTD not provided for day 1 till day 14? In Table 3?
The letter size in Table 7 seems larger than in the other tables.
Discussion:
Pay attention to having spaces between words and carefully check English spelling, vocabular and grammar. The discussion is sometimes hard to read/follow. The authors discuss based on literature data some possible modes of action of HT, but they fail to provide links with their observations … which sometimes provide additional proof and sometimes not.
The conclusion in line 274 that HT+ZnO is positively affecting weight gain is misleading .. even that the authors observed a tendency at d14, no differences are observed at d28 nor regarding daily weight gain.
The authors have used 1000 mg/kg feed HT. Based on what was this concentration chosen?
Line 299 … data are scared??? I guess authors want to say data are scarce ?
Line 300: in broilers HT seems to improve digestability of DM. Do your data substantiate this for pigs? Please elaborate.
Line 304-305 .. improving the area of small intestine for absorption. Do you data substantiate this?
Line 315 etc. what can be deduced from changes in crypt depth? please elaborate?
Can the authors link villus height to enzyme activities? Why investigate amylase and lipase (given the small intestine is not the major site of excretion of these enzymes)?
The others observe some changes in the volatile fatty acids? Could the inclusion of HT and or ZnO have modified the microflora ? if so does the changed profile reflect a possible change? (metabolic profile of the microflora).
Are the increased in IgA, IgM, IgG an indication of an improved immunity ? are they perhaps not indicative of gastrointestinal (or other) problems and thus not necessary a beneficial response? Please provide more arguments for the conclusions in this matter.
Reviewer 2 Report
Generally it is a good and clearly described experiment, however I found there a lot of small editorial errors which I point below.
I have a few considerations for the authors which they should take into account introducing some corrections into manuscript.
The aim of the study is presented very good and it is widely investigated but authors have to be more careful when interpreting results.
l.136 I think that you should describe presented equation: ..."Crdiet× Nutrientfeces ";
add: Crdiet/feaces - chromium in diet/feaces - it is celar, but what is exactly Nutrientfeces/diet - is it total CP, DM, ash and gross energy in feaces/diet?
In context of diarrhea accident authors should confirm that piglets included to the experiment were clinically healthy. That's a pity that blood samples were not taken at the 1st day to confirm it. It will be better to change (line 119) volume of taken blood - 10 ml for 14 days old piglets it is a great volume, especially when later (l.137) only 8 ml is mentioned. And returning to diarrhea: it is a small abuse to write that presented data confirm that HT has a strong anti-diarrhea effect. Generally I agree with it but authors have to write something more. First we don't know the reasons, we can expect that it is connected with transport/locomotion/moving and/or social stress because they appeared in the first period of the experiment - write something about it and later because they had place mostly in control group you can connect it with described IgM concentration - I agree with this interpretation,add something about IgG concentration.
Authors should change a little conclusions. First I don't like term "serum immunity", just HT supplementation improved animals immunity. And because authors didn't infect piglets or to this experiment weren't chosen diarrheal piglets it is difficult in conclusion write that HT supplementation reduced diarrhea rate. In HT group it was 3.5, in ZnO group it was 3.0 and in HT+ZnO group it was 0.33, so better remove it from conclusion.
l 78: - lack of space between words
l 81: - lack of space between words
l 82: "Duroc × (Landrace × Yorkshire)" - in my opinion it is the wrong order, first should be female line, then male one, it should be: (Landrace × Yorkshire) x Duroc
l 89: - lack of space between words
l 121: - lack of space between words
l 147 : - lack of space between words
l 177 : - lack of space between words
l 200 : - lack of space between words
l 202 : - lack of space between words 2 x
l 203 : - lack of space between words
l 204 : - lack of space between words 2 x
l 209 : - lack of space between words
l 238 : - lack of space between words
l 243: - lack of space between words
l 244 : - lack of space between words
l 273 : - lack of space between words 2 x
l 276 : - lack of space between words
l 281 : - lack of space between words
l 284 : - lack of space between words
l 299 : - lack of space between words
l 303 : - lack of space between words
l 308 : - lack of space between words and unnecessary dot
l 316 : - lack of space between words
l 320 : - lack of space between words
l 322 : - lack of space between words
l 323 : - lack of space between words
l 328: is "was duet o bind" should be "was due to..."
l 330 : - lack of space between words
l 332 : - lack of space between words
l 335 : - lack of space between words
l 337 : - lack of space between words
l 344 : - lack of space between words
l 358 : - lack of space between words
Read slowly all references, there is more small errors:
l 386 : is: "Smulikoska SB.;Patuszewska B.;Swiech A.; Ochtabinska.; Mieczkowska A", should be: "Smulikowska S., Pastuszewska B., Święch E , Ochtabińska A., Mieczkowska A., Nguyen VC., Buraczewska L."
l 390 : in title change capital letters into small ones
l 397: should be: Production
l 402 capital letters
Round 2
Reviewer 1 Report
the authors answered to several of the questions in a sufficient manner. however some issues remained. please find below some remarks regarding grammar/spelling and the remaining questions:
line 111: All Ppiglets
line 122-123: Fresh feces collected ... Fresh feces was collected ...
line 158: question: please add the thickness of the sections
line 263: the letter size of the heading 'table 7' seems smaller?
line 307: please check the sentence, there are parts missing: Furthermore, Iin stage ...
line 307: question: what triggered the rise in IgA and IgM ? How could this be beneficial to the piglet? the authors give more information in section lines 370-380 however, regarding IgM the conclusions are not sufficiently substantiated with other reports/references.
line 317: ... during last period. question: what is meant by the 'last period'.
line 317: ... data are scarcely regarding
line 340: question: crypt hyperplasia is not reducing the area available for absorption ... it is the reduction in villus length, which triggers crypt hyperplasia (in order to regenerate the villus), which reduces the area for absorption. Thus the conclusion mentioned in line 344-345 is wrong. If the authors maintain their conclusion, more evidence should be put forward that crypt hyperplasia is a phenomenon that should be prevented? Crypt depth should always be considered along with villus height.
line 349-350: the beneficial effect of tannins on villus height is attributed to its possible effects on the microbiome? (antimicrobial effects, effects on oxidation) how do these affect villus lenght? please provide more details on the mechanistics?
line 380-382: check grammar and letter size
line 489-490: check letter size
